# In Vitro Effects of Cannabidiol on Activated Immune–Inflammatory Pathways in Major Depressive Patients and Healthy Controls

**DOI:** 10.3390/ph15040405

**Published:** 2022-03-26

**Authors:** Muanpetch Rachayon, Ketsupar Jirakran, Pimpayao Sodsai, Siriwan Klinchanhom, Atapol Sughondhabirom, Kitiporn Plaimas, Apichat Suratanee, Michael Maes

**Affiliations:** 1Department of Psychiatry, Faculty of Medicine, Chulalongkorn University and King Chulalongkorn Memorial Hospital, The Thai Red Cross Society, Bangkok 10330, Thailand; muanpetch.mp@gmail.com (M.R.); ket.kett@hotmail.com (K.J.); atapol.s@gmail.com (A.S.); 2Maximizing Thai Children’s Developmental Potential Research Unit, Department of Pediatrics, Faculty of Medicine, Chulalongkorn University, Bangkok 10330, Thailand; 3Center of Excellence in Immunology and Immune-Mediated Diseases, Department of Microbiology, Faculty of Medicine, Chulalongkorn University, Bangkok 10330, Thailand; yokpim@gmail.com; 4Division of Immunology, Department of Microbiology, Faculty of Medicine, Chulalongkorn University, Bangkok 10330, Thailand; siriwanklinchanhom@gmail.com; 5Advanced Virtual and Intelligent Computing (AVIC) Center, Department of Mathematics and Computer Science, Faculty of Science, Chulalongkorn University, Bangkok 10330, Thailand; kplaimas@gmail.com; 6Department of Mathematics, Faculty of Applied Science, King Mongkut’s University of Technology North Bangkok, Bangkok 10800, Thailand; apichat.s@sci.kmutnb.ac.th; 7IMPACT Strategic Research Center, Barwon Health, Geelong, VIC 3220, Australia; 8Department of Psychiatry, Medical University of Plovdiv, 4002 Plovdiv, Bulgaria

**Keywords:** depression, mood disorders, inflammation, neuroimmunomodulation, cytokines, biomarkers, psychiatry

## Abstract

Major depressive disorder and major depressive episodes (MDD/MDE) are characterized by the activation of the immune–inflammatory response system (IRS) and the compensatory immune–regulatory system (CIRS). Cannabidiol (CBD) is a phytocannabinoid isolated from the cannabis plant, which is reported to have antidepressant-like and anti-inflammatory effects. The aim of the present study is to examine the effects of CBD on IRS, CIRS, M1, T helper (Th)-1, Th-2, Th-17, T regulatory (Treg) profiles, and growth factors in depression and healthy controls. Culture supernatant of stimulated (5 μg/mL of PHA and 25 μg/mL of LPS) whole blood of 30 depressed patients and 20 controls was assayed for cytokines using the LUMINEX assay. The effects of three CBD concentrations (0.1 µg/mL, 1 µg/mL, and 10 µg/mL) were examined. Depression was characterized by significantly increased PHA + LPS-stimulated Th-1, Th-2, Th-17, Treg, IRS, CIRS, and neurotoxicity profiles. CBD 0.1 µg/mL did not have any immune effects. CBD 1.0 µg/mL decreased CIRS activities but increased growth factor production, while CBD 10.0 µg/mL suppressed Th-1, Th-17, IRS, CIRS, and a neurotoxicity profile and enhanced T cell growth and growth factor production. CBD 1.0 to 10.0 µg/mL dose-dependently decreased sIL-1RA, IL-8, IL-9, IL-10, IL-13, CCL11, G-CSF, IFN-γ, CCL2, CCL4, and CCL5, and increased IL-1β, IL-4, IL-15, IL-17, GM-CSF, TNF-α, FGF, and VEGF. In summary, in this experiment, there was no beneficial effect of CBD on the activated immune profile of depression and higher CBD concentrations can worsen inflammatory processes.

## 1. Introduction

Activated immune-inflammatory pathways are among the most important pathways involved in the pathophysiology of major depressive disorder (MDD) and a major depressive episode (MDE) in bipolar disorder [1,2,3,4]. Activation of the immune–inflammatory response system (IRS) in MDD/MDE is shown by a pro-inflammatory cytokine profile, including activated M1 macrophage cells, as indicated by elevated levels of interleukin (IL)-1β, IL-6, IL-8 (or CXCL-8), and tumor necrosis factor-α (TNF-α); T helper-1 (Th-1) cells with increased interferon-γ (IFN-γ), IL-2, and IL-12 levels; and activated Th-17 cells with increased IL-17 and IL-6 [1,2]. Depression (MDD/MDE) is not only accompanied by activation of the IRS but also by activation of the compensatory immune-regulatory system (CIRS), which tends to downregulate the primary immune response and hyper-inflammation [1]. Activation of CIRS profiles in MDD/MDE is indicated by activated Th-2 cells with increased IL-4 and IL-5 levels and activated T regulatory (Treg) cells, characterized by increased IL-10, a major negative immunoregulatory cytokine [1].

The imbalance between pro-inflammatory and anti-inflammatory cytokines plays an important role in the onset of depression [1]. Importantly, some cytokines or chemokines, which are increased in MDD/MDE, have neurotoxic effects including IRS cytokines (e.g., IL-1, IL-2, IL-6, IL8, TNF-α, IFN-γ), some chemokines (e.g., CCL1, CCL2, CCL11, CCL5, CXCL8, CXCL10) and even CIRS cytokines (e.g., IL-4) [1]. Therefore, IRS and especially increased neurotoxic cytokines and chemokines, are new drug targets in the treatment of MDD/MDE [5]. Antidepressants of different classes reduce the ex vivo stimulated production of M1 and Th1 cytokines and upregulate that of interleukin-10 [6]. Maes et al. describe that new, more adequate antidepressants should target the activated IRS and associated oxidative and nitrosative stress and neurotrophic pathways [5].

There is emerging evidence that cannabidiol (CBD), a phytocannabinoid that is isolated from the cannabis plant, may have antidepressant activities, as indicated in animal models of depression [7,8,9]. For example, in a rodent model of depression, CBD and imipramine have comparable antidepressant-like effects [10]. There is also evidence that CBD may reduce self-reported depression and anxiety in clinically depressive and anxiety patients [11]. The latter authors reviewed that CBD may improve neurotrophic pathways in the brain, which are reduced in depressive-like behaviors and MDD/MDE, including hippocampal neurogenesis and synaptogenesis, and brain-derived neurotrophic factor (BDNF) signaling in the prefrontal cortex and hippocampus [7]. In addition, CBD has also antioxidant effects [12,13,14].

CBD has a low affinity towards CB_1_ and CB_2_ receptors compared to other cannabinoids such as Δ^9^-tetrahydrocannabinol [15]. 2-arachidonoylglycerol (2AG) and anandamide (AEA) are endocannabinoids that can activate CB_2_ receptors, resulting in different effects on inflammatory processes [12]. For example, 2-AG may promote leukocyte adhesion and release chemokines that promote leukocyte recruitment [12,16]. Endothelial cells cultured with 2-AG for one hour produce TNF-α for 24 h [17]. On the other hand, AEA may decrease the production of cytokines such as IL-4, IL-6, CXCL-8, and IFN-γ [18,19] and may increase the anti-inflammatory cytokine IL-10 [20]. CB_2_ knockout mice show an increased production of pro-inflammatory cytokines, increased leukocyte recruitment and increased tissue damage [18,21,22], while CB_2_ receptor agonists may promote anti-inflammatory effects [14,18,23,24]. The CB_1_ receptor also has immune-modulatory effects, whereby activation of CNS CB_1_ receptors is accompanied by reduced production of reactive oxygen species (ROS) and decreased neuronal tissue damage [14,25]. On the other hand, activation of peripheral CB_1_ receptors may aggravate inflammation via increasing ROS and activation of nuclear factor (NF)-κB [14]. As such, endocannabinoids have complex effects on immune system functions and may promote as well as inhibit inflammatory processes through the CB_2_ receptors. In vivo treatment with a low dose of CBD may decrease TNF-α production in LPS-treated mice [26] and CBD at higher doses highly significantly attenuate the expression of pro-inflammatory cytokines by CD3+ T cells including TNF-α, IL-2, IL-17, IFN-γ and GM-CSF (granulocyte colony-stimulating factor) [27].

Thus, it is plausible that CBD may exert antidepressive effects by reducing M1, Th-1, or Th-17 pathways or increasing CIRS functions (e.g., IL-10 production) thereby reducing neuro-affective toxicity. Nevertheless, so far there is no evidence on whether CBD may modulate the IRS and CIRS and M1, Th-1, Th-2, and Th-17 activities and the production of chemokines and growth factors in patients with MDD/MDE.

Hence, the aim of the study is to examine the effects of CBD on M1, Th-1, Th-2, Th-17, Treg, IRS, and CIRS profiles and cytokines, immune-related neurotoxicity, chemokines, and growth factors. The specific hypotheses are (a) CBD has anti-inflammatory effects by decreasing M1, Th-1, Th-17, and IRS profiles and enhancing Treg and CIRS functions; and (b) CBD attenuates the increased M1, Th-1, Th-17, and IRS responses in depression.

Results are shown as mean ± SD. F: results of analysis of variance; X^2^: analysis of contingency tables, FEPT: Fisher’s exact probability test

BMI: body mass index; HAM-D: Hamilton Depression Rating Scale score; STAI: Spielberger State and Train Anxiety, State version

## 2. Results

### 2.1. Demographic and Clinical Data

Table 1 shows the socio-demographic data and clinical data of the patients and controls in this study. There were no significant differences in sex distribution, education, and smoking between the study groups. Patients were somewhat younger and showed a higher BMI. Nevertheless, all those variables were controlled for by entering these data in the GEE analysis. The mean HAM-D and STAI scores were significantly higher in patients as compared with controls, indicating that most patients suffered from medium to severe clinical depression and anxiety.

### 2.2. Effects of CBD

Table 2 shows the results of GEE analyses with the effects of treatment. CBD 0.1 µg/mL did not have significant effects on the immune profiles. CBD 1.0 µg/mL significantly increased the growth factor and reduced the CIRS profile. At 10.0 µg/mL, CBD significantly decreased Th-1, Th-17, CIRS, and the neurotoxicity profiles but increased the growth factor and T cell growth profiles. The effects of time on these seven immune profiles remained significant after FDR *p* correction (at the *p* = 0.0012 level).

There were no significant effects of any of the drugs on any of the immune profiles (even without FDR *p* correction). For example, there were no significant effects of sertraline (*p* = 0.528), other antidepressants (0.982), benzodiazepines (*p* = 0.944), and atypical antipsychotics (*p* = 0.172) on the Th-1 profile, while the effects of treatment remained significant at *p* < 0.001. In addition, we also examined the interaction between the drug state and the 4 conditions (vehicle and 3 CBD concentrations), but none of these was significant. For example, when examining the Th-1 profile, there were no significant interactions between time × sertraline (*p* = 0.955), other antidepressants (*p* = 0.905), benzodiazepines (*p* = 0.545), and antipsychotics (*p* = 0.608). Moreover, gender, age, BMI, and smoking did not affect any of the immune profiles even without FDR *p* correction.

Table 3 shows the impact of CBD on the different cytokines/growth factors. CBD 0.1 µg/mL had no significant effects on any of the cytokines/growth factors. CBD 1.0 µg/mL significantly decreased the production of sIL-1RA, CXCL8, IL-9, IL-10, IL-13, CCL11, IFN-γ, CCL2, CCL4, and CCL5, and increased the production of IL-1β, IL-2, IL-4, IL-15, FGF, GM-CSF, CXCL10, TNF-α, and VEGF. CBD 10.0 µg/mL had a major impact on most immune markers (except IL-2). CBD 10.0 µ/mL dose-dependently (from CBD 1.0 to 10.0 µg/mL) increased IL-1β, IL-4, IL-15, IL-17, GM-CSF, TNF-α, FGF, VEGF, and decreased sIL-1RA, IL-6, CXCL8, IL-9, IL-10, IL-13, CCL11, G-CSF, IFN-γ, CCL2, CCL3, CCL4, and CCL5. Such a dose-response was not found in the case of IL-2, CXCL10, and PDGF.

### 2.3. Differences Depression Versus Controls

Table 4 shows the results of the same GEE analyses as described in Table 2 but with the focus on the group differences and time X group interactions (the latter are shown only when they are significant). Table 4 shows that the Th-2, Th-17, Th-2, IRS, CIRS, Tcell activation, and neurotoxicity profiles were all higher in depression than in controls, while all interaction terms were non-significant. These intergroup differences remained significant after applying FDR *p* correction (at *p* = 0.0484). Table 5 shows the differences in the separate cytokines/growth factors between both groups: sIL-1RA, IL-5, IL-9, IL-12, IL-15, IL-17, FGF, G-CSF, IFN-γ, CXCL10, TNF-α, and VEGF were significantly higher in depression than in controls. CCL3 was significantly lower in depression than in controls. There were significant interaction patterns for CXCL8, IL-10, GM-CSF, and CCL5. ESF, Figure 1 shows that depression patients have at the 3 CBD conditions lower GM-CSF than controls, whereas in control conditions, there are no differences. ESF, Figure 2 shows that CCL5 levels were higher in patients than in controls in all four conditions. ESF Figure 3 shows the interaction pattern for CXCL8, indicating that this chemokine was higher in depressed patients than in the control, CBD 0.1 and 1.0 µg/mL conditions, whereas those intergroup differences almost disappeared in the CBD 10.0 µg/mL condition. This indicates that the latter concentration tends to normalize the increased CXCL8 values. IL-10 levels were not significantly different between both groups at any condition and there was a trend towards higher levels in the control, 0.1 and 1.0 µg/mL conditions, whilst there was a trend towards lowered levels in the 10.0 µg/mL condition in depression as compared with controls.

### 2.4. Enrichment Analysis in the Effects of CBD on Cytokines

In this analysis, we entered the DEPs which were upregulated and downregulated by 1.0 µg/mL. When performing annotation and enrichment analyses, the same GO terms with large gene members were often found in both the upregulated and downregulated networks. To avoid GO terms with a large member of genes, only enriched GOs with a size of their gene members < 90 genes were of interest. After that, we filtered to select the enriched GOs, which contain our genes > 7% with a gene ratio > 25%. Table 6 and Figure 1 and Figure 2 show the results and the dot plots of the GO enrichment analyses performed on the upregulated and downregulated genes. To delineate the relationship between these two sets of DEPs, we examined the protein interactions. The gene members in the GO:0042531 and GO:0048245 terms were aggregated and included in the up and downregulated DEP sets. All interactions among these genes were retrieved using STRING database version 11.5 [28] with a high confidence score > 0.9. Some genes that had no interactions with this criterion were then discarded from the plot; except VEGFA and IL9 which are in our DEP list. Figure 3 displays the plot of all protein–protein interactions among these genes. VEGFA and IL9 were not found to interact with the other DEPs. IL-4 and CCL5 were present in both GO terms. The results show that CBD increases the regulation of tyrosine phosphorylation of STAT protein and decreases eosinophil chemotaxis. Table 7 shows the REACTOME pathways that are over-represented in the enlarged PPI networks of the upregulated and downregulated DEPs.

## 3. Discussion

### 3.1. Effects of CBD on Immune Profiles and Cytokines/Growth Factors

The first major finding of this study is that CBD 0.1 µg/mL has no significant effects on any of the immune profiles, CBD 1.0 µg/mL decreases CIRS activity and increases growth factor production, and that CBD 10.0 µg/mL significantly suppresses Th-1, Th-17, IRS, CIRS, and neurotoxicity profiles, while increasing the growth factor and T cell growth profiles. Millar et al. [29] in a systematic review showed that the administration of CBD (oral route) used in human studies ranged between <1 to 50 mg/kg/day [29]. The same authors [30] also published another systematic review on the pharmacokinetics of CBD administration and found that administration of 800 mg CBD (oral route) results in plasma concentrations of 0.221 µg/mL (at 3.0 h) and 0.157 µg/mL (at 4.0 h) [30], although other studies yielded a broader concentration range. Moreover, 20 mg CBD intravenously resulted in 0.686 µg/mL 3 min post-administration [30]. Based on those results, we decided to employ 0.1 and 1.0 µg/mL CBD in our ex vivo experiments. Nevertheless, in the rodent, the concentrations in mesenteric lymph nodes were more than 50-fold higher than in the spleen and 250-fold higher than in plasma [27]. The latter authors ascertain that oral CBD coupled with dietary lipids may considerably increase delivery of CBD to the plasma and mesenteric lymphatic system. As such, lymphatic PBMCs may be exposed to high CBD concentrations in the 5–20 µg/mL range when using this targeted approach, which will eventually be applied in patients with inflammatory disorders [27]. Accordingly, in our study, we employed not only the 0.1 and 1.0 µg/mL concentrations but also the 10.0 µg/mL concentration. It is important to note that, in animal studies, there is a significant association between plasma and brain CBD concentrations, whereby a higher intake results in higher plasma and brain concentrations [31,32].

Therefore, we may conclude that CBD treatments, which normally use the low oral dosages, will not result in any significant effects on M1, Th-1, Th-2, Th-17, IRS, CIRS, or the neurotoxic profiles and on the cytokines/growth factors measured here. Nevertheless, when using the higher intravenous dosage, CIRS functions may be suppressed as indicated by significant suppression of IL-10, IL-13, and sIL-1RA production, while the dose-response effects of CBD 1.0 to 10.0 µg/mL indicate a reduced production of all CIRS products, and increased production of IRS products including IL-6, CXCL-8, IL-9, CCL11, G-CSF, IFN-γ, CCL2, CCL3, CCL4, and CCL5.

These findings extend those of previous papers, showing that CBD has a dose-related effect on immune cells [33]. Different studies established that CBD 1.0–10.0 µg/mL may suppress IFN-γ production by PBMCs [34,35,36]. Kozela et al. [35] found that CBD 1 µM had a dose-related inhibitory effect on IL-6 production and Anil et al. [37] reported that CBD 4 µg/mL can reduce CXCL8 levels more than dexamethasone in a model of lung epithelial cells, although treatment with CBD in macrophages may result in increased CXCL8 levels [37]. Zgair et al. [27] observed that the expression of IFN-γ and GM-CSF by CD3+ T cells of multiple sclerosis patients, but not normal controls, was dose-dependently reduced by CBD 2.5–20 µg/mL.

On the other hand, our study established that CBD 1.0 to 10.0 µg/mL dose-dependently increased the production of growth factors including FGF and VEGF, and IL-1β, IL-4, IL-15, IL-17, GM-CSF, and TNF-α. As such, the higher therapeutic dosages and the targeted treatment with lipid-associated CBD may induce pro-inflammatory effects which, linked to the anti-CIRS effects, may have clinical relevance.

Our results extend those of previous papers showing that CBD 1 µM had no effect on TNF- α production while CBD 10 µM induced its production. [38]. On the other hand, an animal model revealed that higher concentrations of CBD (50 µg/mL, 100 µg/mL) can inhibit TNF-α production from microglial cells [39]. CBD 1.0 µg/mL and 10 µg/mL may enhance plasma IL-4 levels, whereas rodent studies showed that CBD 20 mg/kg/day (IV) may reduce plasma IL-4 levels [40]. Other studies established that CBD 1 µM, but not 0.5 µM, can inhibit IL1-β secretion from monocytes [38] and that the dose-response curve of CBD 2.5–20 µg/dL indicated suppressant effects on IL-17, GM-CSF, and TNF-α. In our study, CBD 1.0 µg/mL, but not 10.0 µg/mL, significantly increased IL-2 production. In this respect, Chen et al. reported that CBD may enhance or suppress IL-2, depending on the stimulus condition and that in strongly stimulated cells, CBD may suppress IL-2, but following lower stimuli, CBD may enhance cytokine production [41]. Kaplan et al. investigated the effects of CBD in stimulated splenocytes and splenic T cells and found that CBD 1–10 µM dose-dependently suppresses IL-2 production [34].

The above differences between studies may be explained by the different stimuli and models used to assess the immune network. We used a combination of LPS and PHA to stimulate diluted whole blood and analyzed cytokine and growth factor production in culture supernatant [42]. This diluted whole blood method more adequately reflects the in vivo cytokine production than assays on isolated PBMCs [42]. The method used in the present study retains all-natural cell-to-cell connections, whereas techniques that isolate PBMC subtypes change the lymphocyte/monocyte ratio, thereby affecting cytokine production [42]. Moreover, because some cytokines (including the master immunoregulatory cytokine IL-10) are produced by macrophages, Th-0, Th-1, Th-2, and Treg cells [43], it is critical to assess IL-10 in whole blood cultures with conserved cell-to-cell interactions.

### 3.2. Differences in Immune Profiles in Depression Versus Controls

The second major finding of this study is that depression is accompanied by increased production of the Th-1, Th-2, Th-17, and Treg cytokines and partial activation of the M1 phenotype (increased production of TNF-α, sIL-1RA, and CXCL8) as compared with healthy controls. These findings support the results of a recent review showing that depression is accompanied by increased activities of M1, Th-1, Th-2, Th-17, and Treg cytokines [1].

Regarding the M1 macrophage profile, increased TNF-α, sIL-1RA, and CXCL8 levels were also established in recent meta-analyses [44,45]. However, in the current study, the stimulated production of IL-6 was not elevated in depression, whereas increased serum/plasma IL-6 was established in MDD/MDE in reviews and a meta-analysis [1], while an early study found increased IL-6 in culture supernatant of PBMCs [46]. The current study and a recent meta-analysis [44] did not detect any changes in the stimulated production or serum/plasma levels of IL-1β, whereas an early study reported higher stimulated production of IL-1β in MDD with melancholic features as compared with controls [46]. Köhler et al. also examined CXCL8 but could not find any changes in MDD [44]. While, in the current study, CCL3 was significantly lower in depression, Köhler et al. [44] did not detect any differences in this chemokine. Nevertheless, as explained above, differences among studies may be explained by differences in the media used, e.g., culture supernatant of diluted whole blood versus serum/plasma, and also by differences in depression phenotypes, e.g., melancholia versus simple major depression.

Regarding the Th-1 profile, we found highly significant elevations in culture supernatant IFN-γ and IL-12, but not IL-2, production in MDD/MDE. Some previous studies also reported increased IFN-γ levels in MDD patients [46,47]. Moreover, lowered tryptophan and increased levels of tryptophan catabolites in MDD indicate that indoleamine 2,3-dioxygenase 1 (IDO1) may be stimulated by increased levels of IFN-γ [48]. Daria et al. [49] found that serum IFN-γ levels were decreased in treatment naïve MDD patients as compared with healthy controls, and the Kohler et al. meta-analysis established decreased IFN-γ levels in MDD [44]. Nevertheless, since IFN-γ is difficult to measure in normal serum/plasma, reporting lowered levels in MDD as compared with controls is not adequate. Previous studies found indicants of increased IL-2 mechanisms in MDD, including a higher percentage of patients with measurable IL-2, and increased serum sIL-2R levels and CD25+ bearing T cells) [46], whereas the current study did not detect that IL-2 is increased in culture supernatant of depressed patients. Our results that IL-12 production is increased in depression further supports the finding of the Kohler et al. meta-analysis. Overall, it appears that Th-1 activation is a hallmark of depression whereas M1 macrophages are partially activated. Such findings support the theory of Maes et al. [46] that activated cell-mediated immune pathways including Th-1 activation are the most important hallmarks of depression.

We found significant increases in Th-2 cytokine production with increased levels of IL-5 and IL-9 in MDD patients, whereas there were no differences in IL-4 and IL-13. A previous meta-analysis [44] found that serum/plasma IL-13 levels were significantly elevated in MDD but that IL-4 and IL-5 did not significantly differ between groups. Other authors found increased IL-5 levels in medication-free MDD patients and MDD patients with or without obesity as comorbidity [50,51]. Becerril et al. [52] reported that MDD patients have a higher level of IL-9 compared with healthy controls.

Here, we report that the Th-17 T cell line is activated in depression, as indicated by increased supernatant IL-17 production. Both the review by Maes and Carvalho [1] and the meta-analysis by Kohler et al. [44] showed that IL-17 levels are increased in depression. In the present study, we identified that the IRS as well as CIRS immune profiles were higher in depression than in controls, which supports the review of Maes and Carvalho [1]. However, it should be stressed that the master regulatory cytokine IL-10 was not significantly higher in depression versus controls indicating that those CIRS functions are insufficient to damp IRS activation.

Importantly, in the current study, we found that the neurotoxicity profile was significantly higher in patients than controls, which was attributable to increased CXCL8, IL-12, IL-15, IL-17, TNF-α, IFN-γ, CXCL10, and CCL5 levels. Such findings further support the neurotoxicity theory of both MDD/MDE, which conceptualizes that increased levels of neurotoxic cytokines/chemokines exert neuro-affective toxicity on brain functions leading to depression [1]. These effects are additionally aggravated by increased nitro-oxidative stress, hypernitrosylation, bacterial load, and lower levels of neuroprotective antioxidants [53].

Interestingly, depression was also accompanied by increased levels of growth factors, including FGF, PDGF, and VEGF. Previously, Wu et al., in a meta-analysis, found higher peripheral FGF in MDD as compared with healthy controls [54]. They hypothesized that increases in peripheral FGF2 may be a compensatory effect in order to counter the increased oxidative stress and dysfunctional blood-brain-barrier [54]. Moreover, MDD patients who received treatments had lower levels of peripheral FGF [55]. We also found elevations in PDGF and VEGF levels in MDD/MDE patients which extends the results of previous studies [56,57,58]. Increased PDGF may constitute a compensatory response since exposure to prolonged stress and higher levels of pro-inflammatory cytokines may damage the blood–brain barrier, resulting in increased PDGF production [59,60]. However, Raymond et al. found no differences or even lower VEGF levels in MDD patients as compared with healthy controls [56].

### 3.3. CBD Does Not Normalize the Immune Activation in Depression, on the Contrary

The third major finding of our study is that CBD did not normalize the aberrations in the immune system in patients with MDD/MDE. This contrasts with our a priori hypothesis that CBD may have anti-depressive effects through the regulation of immune-inflammatory processes. Even worse, CBD 1.0–10.0 µg/mL significantly reduced CIRS activity, including suppressant effects on IL-10, sIL-1RA, and IL-13, and increased FGF and VEGF production. However, CBD 10 µg/mL shows a suppressive effect on Th-1, Th-17, CIRS, and the neurotoxic profile, implying that different CBD dosages and different routes of administration may have different immune-modulatory effects.

Moreover, annotation and enrichment analysis showed that 1.0 µg/mL CBD activated tyrosine phosphorylation of STAT protein, which may induce transcription of genes involved in immune cell activation and division and inflammation [61], cytokine signaling, and Toll-Like Receptor signaling, which all play a role in depression. As such, higher CBD concentrations may aggravate the immune pathophysiology of depression (and other inflammatory disease). In addition, the 1.0 µg/mL concentration also decreases eosinophil chemotaxis as well as the phosphatidylinositol signaling system (converting extracellular to intracellular signals and mediating cell metabolism, survival, and proliferation), the forehead box (FoxO) signaling pathway (mediating cell-cycle control and oxidative stress) and the sphingolipid signaling pathways (producing sphingolipids which are involved in signal transduction) (https://reactome.org, (accessed on 2 February 2022)). Moreover, since CBD significantly decreases IL-10 production, we may conclude that treatments with CBD concentrations ranging from 1.0–1.0 µg/mL should be avoided in depression and probably in most immune and autoimmune disorders.

Apart from immune–inflammatory effects, CBD may modulate other pathways associated with depression. Firstly, CBD may modulate 5-HT1A receptors causing anxiolytic-like and anti-depressive effects in rats [62,63]. Second, CBD can inhibit the cortisol axis in a mice model in association with antidepressant-like effects [64]. Third, the administration of CBD induces sustained antidepressant effects which are associated with changes in synaptic plasticity in the prefrontal cortex through stimulation of the BDNF signaling pathway [65].

## 4. Limitations

The results of this study should be interpreted regarding its limitations. According to a reviewer, a first limitation is the use of a healthy control group. Nevertheless, a control group was included to examine the differences in immune profiles between patients and controls and to examine whether CBD may normalize the immune profile of depressed patients. A second limitation according to a referee is the use of numerous statistical tests “that questions the level of alpha risk”. Nevertheless, the multiple effects of CBD treatment (or depression) on the primary outcome variables (namely the immune profiles) were *p*-corrected for FDR [66]. Moreover, this FDR correction may be too conservative because all profiles are strongly correlated with each other and especially because (as shown in the PPI analyses) cytokine/growth factors function in tightly connected networks and, therefore, cannot be regarded as independent entities. Third, the BMI was higher in depressed patients, and it is known that increasing BMI is accompanied by increases in inflammatory biomarkers [67]. The differences in inflammatory biomarkers between people with a normal BMI and those with overweight are very small, although significant and detectable in large-scale studies [67]. Nevertheless, in the present study, we statistically controlled for any effects of BMI (and age and sex) and found that all results shown in this study did not change after adjusting for BMI, while BMI had no significant effects. By inference, depression explains a much greater part of the variance in the immune tests than BMI, and the effects of the latter may not always be detected in smaller *n* studies. Future research should examine the in vitro effects of CBD on immune profiles in other psychiatric samples which are characterized by immune activation, including schizophrenia.

## 5. Materials and Methods

### 5.1. Participants

We included 30 depressed outpatients recruited at the outpatient clinic of the Department of Psychiatry, King Chulalongkorn Memorial Hospital, Bangkok, Thailand. We included participants of both sexes, aged 18 to 65 years old. They were diagnosed as suffering from MDD or MDE according to DSM-5 criteria and showed moderate to severe depression as assessed with the Hamilton Depression Rating Scale (HAM-D). The healthy volunteers were both males and females in the same age range as the patients. They were recruited by word of mouth from the same catchment area as the patients, namely Bangkok, Thailand. Exclusion criteria for MDD/MDE patients were other DSM-5 axis 1 disorders including schizophrenia, schizoaffective disorders, obsessive-compulsive disorder, post-traumatic stress disorder, psycho-organic disorders, and substance abuse disorders. Exclusion criteria for healthy volunteers were a diagnosis of any axis 1 DSM-5 disorder and a positive family history of MDD and bipolar disorder. Exclusion criteria for both patients and controls were (a) allergic or inflammatory responses three months before the study; (b) neuroinflammatory, neurodegenerative or neurological disorders including epilepsy, Alzheimer’s disease, multiple sclerosis, and Parkinson’s disease; (c) (auto)immune or allergic diseases including chronic obstructive pulmonary disease, inflammatory bowel disease, psoriasis, diabetes type 1, asthma, cancer, inflammatory diseases, and rheumatoid arthritis, (d) a lifetime history of immunomodulatory drugs treatments, (e) recent treatments (three months prior to the study) with therapeutic doses of antioxidants or omega-3 polyunsaturated fatty acid supplements, (f) use of anti-inflammatory drugs (NSAID, steroids) one month prior to the study, and (g) lactating or pregnant women. Part of the patients took psychotropic medication, namely sertraline (*n* = 18), other antidepressants (*n* = 8, including fluoxetine, venlafaxine, escitalopram, bupropion and mirtazapine), benzodiazepines *n* = 22; atypical antipsychotics *n* = 14, and mood stabilizers (*n* = 4). Putative effects of these drug state variables were statistically controlled for.

All controls and patients provided written informed consent prior to participation in this study. The study was conducted according to International and Thai ethics and privacy laws. Approval for the study was obtained from the Institutional Review Board of the Faculty of Medicine, Chulalongkorn University, Bangkok, Thailand (#528/63), which is in compliance with the International Guidelines for Human Research protection as required by the Declaration of Helsinki, The Belmont Report, CIOMS Guideline and International Conference on Harmonization in Good Clinical Practice (ICH-GCP).

### 5.2. Clinical Measurements

Semi-structured interviews were conducted by a research assistant specializing in mood disorders. We used the HAM-D, 17 item version, administered by an experienced psychiatrist to measure the severity of depressive symptoms [68]. The State–Trait Anxiety Inventory (STAI) Thai state version is a psychological inventory that is used to measure the severity of state anxiety [69]. The Mini-International Neuropsychiatric Interview (M.I.N.I.) was used to assess psychiatric axis-1 diagnoses [70].

### 5.3. Assays

After an overnight fast (at least 10 h), 20 mL blood was sampled at 8.00 a.m. and collected in BD Vacutainer^®^ EDTA (10 mL) and BD Vacutainer^®^ SST™ (5 mL) tubes (BD Biosciences, Franklin Lakes, NJ, USA). The SST blood was kept at room temperature (30 min) to allow clotting and to collect serum. Both tubes were centrifuged at 1100× *g* (4 °C, 10 min) and serum or plasma was aliquoted and kept at −80 °C. The natural CBD (99.89% isolate) stock preparation was produced by Love Hemp Ltd., United Kingdom (batch 8406) and its quality was guaranteed by the Medical Cannabis Research Institute, College of Pharmacy, RSU, Bangkok, Thailand.

Whole blood cultures were used for the assay of IL-1β, sIL-1RA, IL-2, IL-4, IL-5, IL-6, IL-7, CXCL8, IL-9, IL-10, IL-12, IL-13, IL-15, IL-17, CCL10, FGF, GM-CSF, IFN-γ, CXCL10, CCL2, CCL3, PDGF, CCL4, CCL5, TNF-α, VEGF, and G-CSF. Appendix A shows a list with the names, abbreviations, and official gene symbols of all cytokines/chemokines/growth factors assayed in the present study. Appendix A shows a list of the different immune profiles examined in the current study.

The effects of CBD on these cytokines/growth factors were studied by stimulating whole blood with PHA and LPS and analyzing the levels of cytokine/growth factor production in culture supernatant. RPMI-1640 medium (Gibco Life Technologies, Carlsbad, CA, USA) with L-glutamine and phenol red and containing 1% of penicillin (Gibco Life Technologies, Carlsbad, CA, USA) were employed with (stimulated) or without (unstimulated) 5 μg/mL PHA (Merck, Germany) and 25 μg/mL lipopolysaccharide (LPS.; Merck, Germany). 1.8 mL of either one of these two media was placed into 24-well sterile plates with 0.2 mL of whole blood, 1/10 diluted. Whole blood was seeded in the 24-well culture plates with CBD, which dissolved with dimethyl sulfoxide (DMSO) to make 1 mg/mL of stock and kept at −80 °C. For each subject, the specimens could be divided into four conditions, namely (1) stimulated with LPS + PHA and incubated 72 h, and (2–4) stimulated with LPS + PHA and incubated 72 h with three different concentrations of CBD, namely CBD 0.1 µg/mL, 1 µg/mL and 10.0 g/mL. Samples were incubated for 72 h in a humidified atmosphere at 37 °C, 5% CO_2_. After incubation, the plates were centrifuged at 1500 rpm for 8 min. Supernatants were taken off carefully under sterile conditions, divided into Eppendorf tubes, and frozen immediately at −70 °C until thawed for assay of the cytokines/growth factors in culture supernatant. These CBD concentrations were chosen based on prior research in humans showing therapeutic CBD serum/plasma levels in the 0.1–1.0 µg/dL range and additionally a 10-fold higher dose (10.0 µg/dL), which is obtained in the intestinal lymphatic system after more targeted treatment [27].

The cytokines were measured using a LUMINEX assay (BioRad, Hercules, CA, USA), which is a bead-based multiplex method. In brief, supernatants were fourfold diluted with media and incubated with coupled magnetic beads for 30 min. Detection antibodies were then added followed by streptavidin-PE for 30 min and 10 min, respectively, before measuring the fluorescence intensities (FI) using the Luminex 200 machine. In the present study, we used the (blank analyte subtracted) FI values in the statistical analysis because FI are generally a more appropriate choice than absolute concentration especially when multiple plates are used [71]. Table 1 shows the range of the FI values that fall within the concentration curve. All samples of all cytokines were well measurable, except IL-7, which showed too many values below the sensitivity of the assay and was, therefore, excluded. IL-13 showed the acceptable limit of 30% values lower than the sensitivity and could be included in the statistical analysis. The inter-assay CV values for all analyses are less than 11%.

### 5.4. Statistical Analysis

Chi-square tests were performed to determine relationships between categorical variables, while analysis of variance (ANOVA) was employed to compare scale variables across categories. We employed generalized estimating equations (GEE) analysis, repeated measures, to determine the impact of CBD treatment on the cytokine profiles and all cytokines/growth factors separately. The pre-specified GEE analyses, repeated measures (unstructured working correlation matrix, a linear scale response, and maximum likelihood estimate as scale parameter method) included fixed categorical effects of time (treatment with vehicle and 3 CBD concentrations), groups (depression versus controls) and time-by-treatment interaction, sex, smoking, and continuous fixed covariates, namely age and BMI. Moreover, the unstimulated values of the respective immune profiles and separate cytokines/growth factors were used as additional covariates but this did not change the results (because there were no significant differences in the unstimulated values between time and groups; results not shown). The immune profiles were the primary outcome variables and if these yielded significant outcomes we also examined the separate cytokines/growth factors. The multiple effects of time or groups on the immune profiles were subjected to false discovery rate (FDR) *p* correction [66]. Moreover, we also entered the drug state, age, sex, BMI, and smoking in the GEE analysis as additional predictors to rule out any effects of these possible confounders. The GEE technique enables us to account for important interactions and confounders when analyzing treatment effects at the subject level while avoiding biased imputations caused by incomplete evaluations. Nevertheless, there were no missing values in any of the demographic, clinical or cytokines/growth factor data (except IL-7, which was deleted from the analyses) assessed in this study. We computed the estimated marginal mean values of the treatment and diagnostic groups and the treatment X group interactions and used (protected) pairwise contrasts (least significant difference at *p* = 0.05) to examine differences between the treatment conditions (CBD compared with vehicle) and the group × treatment interactions. The tests were two-tailed, and statistical significance was defined as *p* < 0.05. IBM SPSS windows version 28 was used to perform the statistical analyses. The statistical studies followed the International Conference on Harmonisation’s E9 statistical guidelines (November 2005). Using a two-tailed test with a significance level of 0.05 and assuming a power of 0.80, an effect size of 0.2, two groups, four measurements with intercorrelations of around 0.4 showed that the estimated sample size for a repeated measurement design ANOVA would be around 44.

### 5.5. Protein-Protein (PPI) Network and GO Enrichment Analysis

GO (biological processes, geneontology.org, accessed on 22 January 2022) enrichment analyses were performed separately on the upregulated and downregulated differentially expressed proteins (DEPs) using the R package clusterProfiler [72] and GO.db [Caslson, M. GO.db: A set of annotation maps describing the entire Gene Ontology. R package version 3.8.2., version 3.8.2.; 2019] [73]. The dot plots and the results of the enrichment analysis are always shown using FDR corrected *p*-values and q-values. The PPI was constructed based on our DEP list and the gene members of the obtained GO terms. All interactions among the DEPs were retrieved from STRING database version 11.5 [28] using a high confidence score > 0.900. In addition, we constructed an expanded PPI network around the upregulated DEPs using OmicsNet 2.0 (OmicsNet) and the IntAct Molecular Interaction Database (https://www.ebi.ac.uk/intact/ (accessed on 2 February 2022) and examined the PPI network for its enrichment terms using the REACTOME (European Bio-Informatics Institute Pathway Database) pathways (https://reactome.org, accessed on 19 September 2021).

## 6. Conclusions

CBD does not normalize the activated immunological profiles associated with depression and, in the higher concentration range, may even worsen inflammatory responses and impair CIRS functions.

## Figures and Tables

**Figure 1 pharmaceuticals-15-00405-f001:**
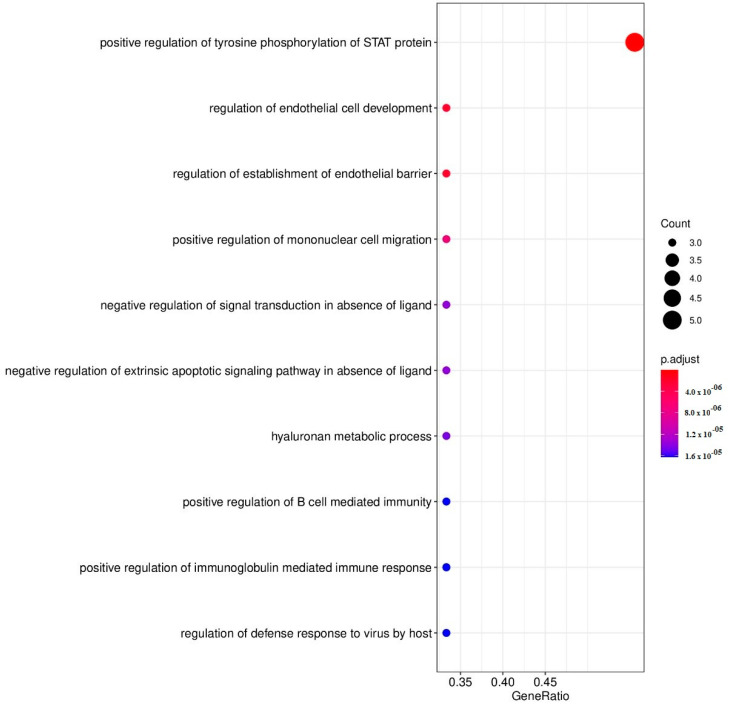
Dot plots of enriched GO functions representing upregulated genes by cannabidiol (1.0 µg/mL).

**Figure 2 pharmaceuticals-15-00405-f002:**
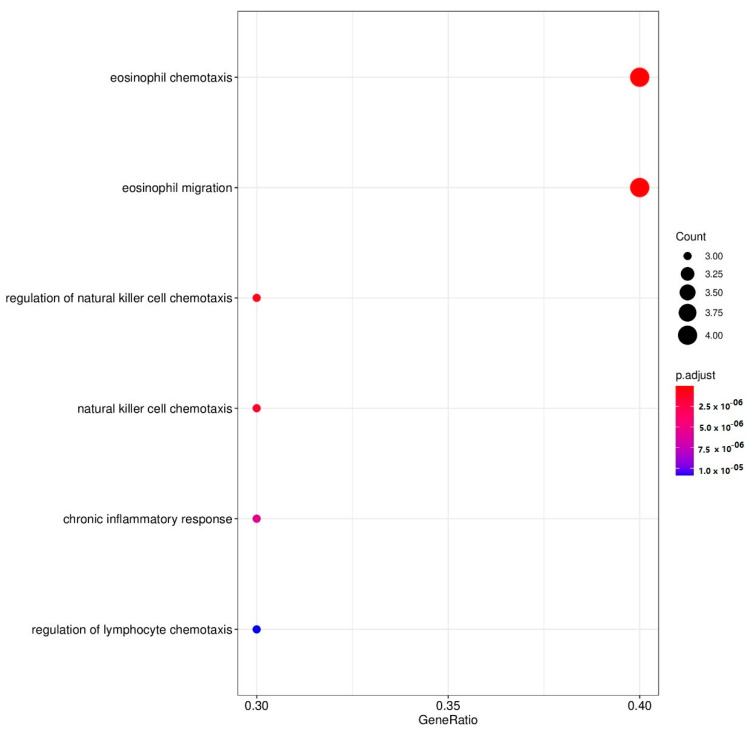
Dot plots of enriched GO functions representing downregulated genes by cannabidiol (1.0 µg/mL). *p* adjust: color intensity indicates variable values of *p* values.

**Figure 3 pharmaceuticals-15-00405-f003:**
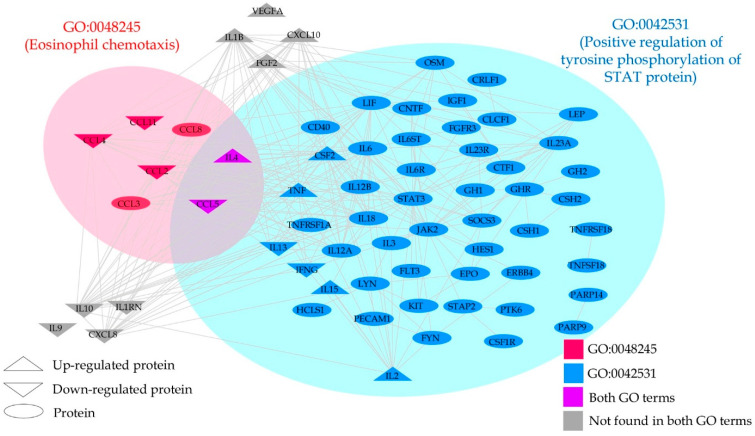
Protein–protein interaction network of high-confidence interactions of the upregulated and downregulated genes as well as gene members of GO:0042531 (positive regulation of phosphorylation of STAT protein) and GO:0048245 (eosinophil chemotaxis).

**Table 1 pharmaceuticals-15-00405-t001:** Demographic and clinical data of the depressed patients and healthy controls (HC) included in the present study.

Variables	HC (*n* = 20)	Depression (*n* = 30)	F/X^2^	df	*p*
Sex (Male/Female)	6/14	11/19	0.24	1	0.626
Age (years)	33.6 (8.0)	28.7 (8.6)	4.28	1/48	0.04
Education (years)	16.1 (2.2)	15.6 (1.4)	0.84	1/48	0.363
BMI (kg/m^2^)	21.3 (2.5)	25.5 (5.9)	8.82	1/48	0.005
HAM-D	0.95 (1.59)	23.5 (5.8)	281.87	1/48	<0.001
STAI	37.7 (10.6)	56.8 (7.2)	57.15	1/48	<0.001
Smoking (N/Y)	18/2	23/7	FEPT	-	0.285

**Table 2 pharmaceuticals-15-00405-t002:** Effects of cannabidiol (CBD) on lipopolysaccharide and phytohaemagglutinin-induced changes in various immune profiles.

Variables (z Scores)	Control ^a^	CBD 0.1 µg/mL ^b^	CBD 1.0 µg/mL ^c^	CBD 10 µg/mL ^d^	Wald	df	*p*
M1	−0.094 (0.122)	−0.079 (0.143)	−0.029 (0.118)	0.114 (0.118)	5.02	3	0.171
Th-1	0.045 (0.115) ^d^	0.013 (0.125) ^c,d^	0.091 (0.121) ^b,d^	−0.374 (0.110) ^a,^^b,c^	53.45	3	<0.001
Th-17	0.018 (0.121) ^d^	−0.013 (0.128) ^d^	0.012 (0.121) ^d^	−0.216 (0.124) ^a,^^b,c^	17.66	3	0.001
Th-2	−0.038 (0.126)	−0.037 (0.123)	−0.075 (0.125)	−0.099 (0.119)	0.96	3	0.812
IRS	−0.008 (0.113)	−0.116 (0.117)	0.004 (0.110)	−0.228 (0.106)	22.53	3	<0.001
CIRS	0.084 (0.131) ^c,d^	0.074 (0.133) ^c,d^	−0.001 (0.126) ^a,b,d^	−0.356 (0.104) ^a,b,c^	37.23	3	<0.001
Tcell	−0.150 (0.113) ^d^	−0.165 (0.112) ^d^	−0.114 (0.099) ^d^	0.226 (0.108) ^a,b,c^	36.30	3	<0.001
GF	−0.139 (0.109) ^c,d^	−0.135 (0.107) ^c,d^	−0.107 (0.107) ^a,b,d^	0.094 (0.117) ^a,b,c^	155.52	3	<0.001
NT	0.052 (0.117) ^d^	0.018 (0.129) ^d^	0.043 (0.122) ^d^	−0.303 (0.116) ^a,b,c^	35.84	3	<0.001

Results of GEE analyses with immune profiles as dependent variables and time, group (depression versus controls) and time by group interactions as explanatory variables, and age, sex, and body mass index as covariates. Shown are the time effects (Wald) with ^a,b,c,d^ indicating pairwise comparisons among the treatment conditions. All data are shown as estimated marginal means (mean ± SE). See ESF Table 2 for explanation of the profiles and cytokines measured in this study. IRS: immune-inflammatory response system, CIRS: compensatory immunoregulatory response system, Tcell: T cell growth, GF: growth factors, NT: neurotoxicity.

**Table 3 pharmaceuticals-15-00405-t003:** Effects of cannabidiol (CBD) on lipopolysaccharide and phytohaemagglutinin-induced changes in various cytokines/chemokines.

Variables z Scores	Control ^a^	CBD 0.1 µg/mL ^b^	CBD 1 µg/mL ^c^	CBD 10 µg/mL ^d^	Wald ×2 (df = 3)	*p* Value
IL-1β	−0.461(0.090) ^c,d^	−0.429 (0.092) ^c,d^	−0.230 (0.096) ^a,b,d^	1.144 (0.123) ^a,b,c^	425.40	<0.001
IL−1RA	0.018 (0.090) ^c,d^	0.005 (0.089) ^d^	−0.007 (0.090) ^a,d^	−0.230 (0.092) ^a,b,c^	168.27	<0.001
IL-2	−0.109 (0.124) ^c^	−0.105 (0.136) ^c^	0.103 (0.134) ^a,b^	−0.018 (0.130)	29.23	<0.001
IL-4	−0.272 (0.125) ^c,d^	−0.227 (0.142) ^c,d^	−0.096 (0.137) ^a,b,d^	0.551 (0.137) ^a,b,c^	61.81	<0.001
IL-6	0.167 (0.130) ^d^	0.152 (0.140) ^d^	0.124 (0.125) ^d^	−0.484 (0.118) ^a,b,c^	105.82	<0.001
CXCL8	0.091 (0.127) ^c,d^	0.091 (0.142) ^d^	−0.007 (0.108) ^a,d^	−0.383 (0.021) ^a,b,c^	21.10	<0.001
IL-9	−0.026 (0.116) ^c,d^	−0.038 (0.114) ^c,d^	−0.072 (0.109) ^a,b,d^	−0.137 (0.107) ^a,b,c^	49.68	<0.001
IL-10	0.433 (0.139) ^c,d^	0.384 (0.132) ^c,d^	0.119 (0.114) ^a,b,d^	−0.992 (0.022) ^a,b,c^	138.95	<0.001
IL-13	0.266 (0.143) ^c,d^	0.246 (0.121) ^d^	0.043 (0.133) ^a,d^	−0.609 (0.079) ^a,b,c^	56.32	<0.001
IL-15	−0.133 (0.095) ^c,d^	−0.133 (0.099) ^c,d^	−0.093 (0.098) ^a,b,d^	0.056 (0.096) ^a,b,c^	125.91	<0.001
IL−17	−0.140 (0.106) ^d^	−0.175 (0.108) ^c,d^	−0.107 (0.110) ^b,d^	0.146 (0.127) ^a,b,c^	36.24	<0.001
CCL11	0.127 (0.127) ^c,d^	0.079 (0.135) ^d^	0.023 (0.135) ^a,d^	−0.132 (0.110) ^a,b,c^	17.00	<0.001
FGF	−0.179 (0.107) ^c,d^	−0.177 (0.105) ^c,d^	−0.121 (0.106) ^a,b,d^	0.208 (0.101) ^a,b,c^	143.85	<0.001
G-CSF	0.046 (0.111) ^d^	0.053 (0.114) ^d^	0.034 (0.115) ^d^	−0.148 (0.100) ^a,b,c^	29.09	<0.001
GM-CSF	−0.168 (0.137) ^c,d^	−0.133 (0.121) ^d^	−0.031 (0.122) ^a,d^	0.422 (0.132) ^a,b,c^	31.88	<0.001
IFN-γ	0.277 (0.123) ^c,d^	0.251 (0.126) ^c,d^	0.196 (0.126) ^a,b,d^	−0.895 (0.071) ^a,b,c^	166.03	<0.001
CXCL10	−0.030 (0.089) ^c,d^	−0.034 (0.090) ^c,d^	0.009 (0.089) ^a,b,d^	−0.183 (0.117) ^a,b,c^	36.70	<0.001
CCL2	0.399 (0.147) ^c,d^	0.355 (0.143) ^c,d^	0.139 (0.140) ^a,b,d^	−0.819 (0.060) ^a,b,c^	159.52	<0.001
CCL3	0.079 (0.109) ^d^	0.079 (0.109) ^d^	0.077 (0.109) ^d^	0.054 (0.112) ^a,b,c^	35.39	<0.001
PDGF	−0.034 (0.030) ^d^	−0.036 (0.026) ^d^	−0.037 (0.09) ^d^	0.098 (0.043) ^a,b,c^	14.77	0.002
CCL4	0.168 (0.128) ^c,d^	0.147 (0.127) ^d^	0.120 (0.123) ^a,d^	−0.316 (0.117) ^a,b,c^	231.01	<0.001
CCL5	0.041 (0.094) ^c,d^	0.023 (0.096) ^c,^^d^	−0.038 (0.096) ^a,b,d^	−0.174 (0.083) ^a,b,c^	64.97	<0.001
TNF-α	−0.048 (0.091) ^c,d^	−0.041 (0.097) ^c,d^	−0.005 (0.099) ^a,b,d^	0.070 (0.091) ^a,b,c^	13.30	0.004
VEGF	−0.124 (0.099) ^c,d^	−0.111 (0.101) ^c,d^	−0.084 (0.099) ^a,b,d^	0.042 (0.103) ^a,b,c^	138.92	<0.001

Results of GEE analyses with cytokines/chemokines as dependent variables and time, group (depression versus controls), and time by group interactions as explanatory variables. Shown are the time effects (Wald) with ^a,b,c,d^, indicating pairwise comparisons among the treatment conditions. All data are shown as estimated marginal means (mean ± SE). See ESF Table 2 for explanation of the profiles and cytokines measured in this study.

**Table 4 pharmaceuticals-15-00405-t004:** Differences in lipopolysaccharide and phytohaemagglutinin-induced changes in various immune profiles between patients with clinical depression and healthy controls (HC).

Variables z Values	HC	Depression	Wald (df = 1)	*p*
M1	−0.131 (0.116)	0.087 (0.205)	0.86	0.354
Th-1	−0.337 (0.119)	0.225 (0.198)	5.59	0.018
Th-17	−0.299 (0.122)	0.199 (0.204)	4.43	0.035
Th-2	−0.374 (0.125)	0.250 (0.196)	7.16	0.007
IRS	−0.366 (0.114)	0.244 (0.184)	7.91	0.005
CIRS	−0.299 (0.169)	0.199 (0.172)	4.18	0.041
Tcell	−0.305 (0.110)	0.203 (0.169)	6.64	0.011
GF	−0.430 (0.029)	0.287 (0.217)	10.71	0.001
NT	−0.286 (0.125)	0.190 (0.197)	4.09	0.043

Results of GEE analyses (df = 1) with immune profiles as dependent variables and time, group (depression versus controls), and time by group interactions as explanatory variables. Only the effects of groups are shown as the interactions between group and time were always non-significant (the time effects are shown in Table 2). All data are shown as estimated marginal means (mean ± SE). See ESF Table 2 for explanation of the profiles and cytokines measured in this study.

**Table 5 pharmaceuticals-15-00405-t005:** Differences in lipopolysaccharide and phytohaemagglutinin-induced changes in various cytokines/growth factors between patients with clinical depression and healthy controls (HC).

Variables z Scores	HC	Depression	Effect	Wald	Df	*p*
IL-1RA	−0.321 (0.106)	0.214 (0.141)	G	9.45	1	0.002
IL-5	−0.442 (0.034)	0.295 (0.217)	G	11.02	1	0.001
CXCL8	−0.312 (0.015)	0.208 (0.192)	G G × T	7.30 8.85	1 3	0.007 0.031
IL-9	−0.178 (0.064)	0.118 (0.121)	G	5.19	1	0.023
IL-10	−0.083 (0.150)	0.055 (0.134)	G × T	13.34	3	0.004
IL-12	−0.391 (0.067)	0.261 (0.204)	G	8.09	1	0.004
IL-15	−0.455 (0.098)	0.303 (0.178)	G	12.81	1	<0.001
IL-17	−0.415 (0.073)	0.277 (0.208)	G	9.71	1	0.002
FGF	−0.404 (0.093)	0.269 (0.183)	G	10.75	1	0.001
G-CSF	−0.421 (0.013)	0.281 (0.219)	G	10.28	1	0.001
GM-CSF	0.134 (0.166)	−0.089 (0.181)	G × T	8.05	3	0.045
IFN-γ	−0.255 (0.145)	0.170 (0.155)	G	4.05	1	0.044
CXCL10	−0.357 (0.140)	0.238 (0.129)	G	9.48	1	0.002
CCL3	0.437 (0.009)	−0.292 (0.220)	G	10.91	1	0.001
CCL5	−0.223 (0.115)	0.149 (0.140)	G G × T	4.21 8.11	1 3	0.040 0.044
TNF-α	−0.246 (0.092)	0.169 (0.148)	G	6.19	1	0.013
VEGF	−0.415 (0.112)	0.277 (0.163)	G	12.54	1	<0.001

Results of GEE analyses with cytokines/growth factors as dependent variables and time, group (depression versus controls), and group by time interactions (G × T) as explanatory variables. The effects of groups and/or the interactions between group and time are shown when significant (the time effects are shown in Table 3). All data are shown as estimated marginal mean (mean ± SE) values.

**Table 6 pharmaceuticals-15-00405-t006:** Results of GO enrichment analysis of biological processes performed on the upregulated and downregulated protein–protein interaction networks of the differently expressed proteins (DEPs) induced by cannabidiol (CBD, 1 µg/mL).

ID	Description (Upregulated DEPs by CBD)	Gene Members	Gene Ratio	*p*-Adjust	*q*-Value
GO:0042531	Positive regulation of tyrosine phosphorylation of STAT protein	71	5/9	2.95 × 10^−8^	6.33 × 10^−9^
GO:1901550	Regulation of endothelial cell development	15	3/9	1.90 × 10^−6^	4.08 × 10^−7^
GO:1903140	Regulation of establishment of endothelial barrier	15	3/9	1.90 × 10^−6^	4.08 × 10^−7^
GO:0071677	Positive regulation of mononuclear cell migration	25	3/9	6.56 × 10^−6^	1.41 × 10^−6^
GO:1901099	Negative regulation of signal transduction in absence of ligand	36	3/9	1.37 × 10^−5^	2.95 × 10^−6^
GO:2001240	Negative regulation of extrinsic apoptotic signaling pathway in absence of ligand	36	3/9	1.37 × 10^−5^	2.95 × 10^−6^
GO:0030212	Hyaluronan metabolic process	37	3/9	1.45 × 10^−5^	3.13 × 10^−6^
GO:0002714	Positive regulation of B cell-mediated immunity	40	3/9	1.62 × 10^−5^	3.48 × 10^−6^
GO:0002891	Positive regulation of immunoglobulin mediated immune response	40	3/9	1.62 × 10^−5^	3.48 × 10^−6^
GO:0050691	Regulation of defense response to virus by host	40	3/9	1.62 × 10^−5^	3.48 × 10^−6^
**ID**	**Description (Downregulated DEPs by CBD)**	**Gene Members**	**Gene Ratio**	***p*-Adjust**	***q*-Value**
GO:0048245	Eosinophil chemotaxis	19	4/10	4.82 × 10^−8^	1.54 × 10^−8^
GO:0072677	Eosinophil migration	23	4/10	7.60 × 10^−8^	2.42 × 10^−8^
GO:2000501	Regulation of natural killer cell chemotaxis	9	3/10	6.95 × 10^−7^	2.21 × 10^−7^
GO:0035747	Natural killer cell chemotaxis	11	3/10	1.15 × 10^−6^	3.66 × 10^−7^
GO:0002544	Chronic inflammatory response	20	3/10	5.37 × 10^−6^	1.71 × 10^−6^
GO:1901623	Regulation of lymphocyte chemotaxis	27	3/10	1.09 × 10^−5^	3.46 × 10^−6^

**Table 7 pharmaceuticals-15-00405-t007:** REACTOME pathway classifications of the differently expressed proteins (DEPs) in the enlarged protein–protein interaction networks of the upregulated and downregulated DEPs induced by cannabidiol (1.0 µg/mL).

Pathways in Upregulated DEPs	Total	Expected	Hits	*p*	FDR *p*
Cytokine Signaling in Immune system	286	2.18	18	1.08 × 10^−12^	1.51 × 10^−9^
TRIF-mediated TLR3/TLR4 signaling	87	0.665	10	6.88 × 10^−10^	2.70 × 10^−7^
MyD88-independent cascade	88	0.672	10	7.71 × 10^−10^	2.70 × 10^−7^
Toll Like Receptor 3 (TLR3) Cascade	88	0.672	10	7.71 × 10^−10^	2.70 × 10^−7^
Activated TLR4 signaling	100	0.764	10	2.76 × 10^−9^	7.75 × 10^−7^
**Pathways in Downregulated DEPs**					
Phosphatidylinositol signaling system	100	0.694	31	1.59 × 10^−46^	5.33 × 10^−44^
FoxO signaling pathway	294	2.04	38	1.53 × 10^−42^	2.57 × 10^−40^
Sphingolipid signaling pathway	189	1.31	29	2.06 × 10^−33^	2.30 × 10^−31^

## Data Availability

Data is contained within the article and Appendix A.

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
