# Peer review of "In Vitro Effects of Cannabidiol on Activated Immune–Inflammatory Pathways in Major Depressive Patients and Healthy Controls"

_pharmaceuticals, 2022, doi:10.3390/ph15040405_

Round 1

Reviewer 1 Report

Interesting study. One limitation is the use of a  healthy control group.  For some data, to infere the results to depression the authors may use a psychiatric group except depression. A second limitation is the use of numerous statistical tests that questions the level of alpha risk. This point must be discussed. Is a Bonferroni correction usefull?

A minor point. Materiel and methods must be presented after the results (see authors instruction).

Author Response

Interesting study. One limitation is the use of a healthy control group For some data, to infere the results to depression the authors may use a psychiatric group except depression.

@ANSWER: This is addressed in a new section, namely Limitations. In reads:

The results of this study should be interpreted with regard to the limitations. According to a reviewer, a first limitation is the use of a healthy control group. Nevertheless, a control group was included to examine the differences in immune profiles between patients and controls and to examine whether CBD may normalize the immune profile.

and

Future research should examine the in vitro effects of CBD on immune profiles in other psychiatric samples which are characterized by immune activation, including schizophrenia.

 A second limitation is the use of numerous statistical tests that questions the level of alpha risk. This point must be discussed. Is a Bonferroni correction usefull?

@ANSWER: addressed in the tekst as:

A second limitation according to the referees is the use of numerous statistical tests that questions the level of alpha risk. Nevertheless, the multiple effects of CBD treatment (or depression) on the primary outcome variables (namely the immune profiles) were p-corrected for FDR [30]. Moreover, this FDR correction may be too conservative because all profiles are strongly correlated with each other and especially because (as shown in the PPI analyses) cytokine/growth factors function in a tightly connected network and therefore cannot be regarded as independent entities.

A minor point. Materiel and methods must be presented after the results (see authors instruction).

@ANSWER: we presented the methods part after the discussion

Reviewer 2 Report

Dear Authors,

Your manuscript contains very interesting and valuable data regarding the link between major depressive disorder and inflammation. The presence of neuroinflammation has been reported in depressive patients, but treatment has yet to be developed. CBD, on the other hand, has been proposed as a highly promising drug candidate for a wide variety of CNS disorders. While this study might be of great interest to readers, it needs major revision before publication.

First, the title should be revised because in the current form it points toward a clinical study conducted with CBD. In contrast, the study was performed on whole blood culture.

Second, the abstract should be rewritten:
- Methods, Results, Conclusion words should be omitted
- the Results part should be rephrased because the observed increase of Th-1, Th-2 etc. profile has been induced by lipopolysaccharide + phytohaemagglutinin, and is not an inherent characteristic of depression. 
- Conclusion: there was no beneficial effect of CBD on the activated immune profile - the authors should emphasize this, and not speculate "very complex". And again, this should not be generalized - please specify that in this experimental context.

Third, in the Introduction section, more evidence exists that CBD might have antidepressant-like effects. In lines 82-84, please cite 10.3390/biom10050801 and 10.1016/j.pnpbp.2021.110508 to have more updated reference list.

Fourth, in the Methods section:
The authors did not set any criteria for BMI when selecting patients for the study. Moreover, in the Results section, they showed that the depressive group has had a significantly higher BMI. However, there is plenty of evidence that obesity can be linked with a low-grade chronic inflammation characterized by macrophage activation and TNFa production. It is stated that this confounding was controlled for in the statistical analysis phase, but it might be worth commenting on this.

Minor comments:
Results

Table 4 can be compacted by eliminating the df column and including the data in the text of the table caption. Tables 6 and 7 should be formatted according to the journal's guidelines. 

Discussion

Line 440 - This reference is more relevant: 10.3390/ijms23010094

Line 585 - Please check.

Author Response

REFEREE #2

Your manuscript contains very interesting and valuable data regarding the link between major depressive disorder and inflammation. The presence of neuroinflammation has been reported in depressive patients, but treatment has yet to be developed. CBD, on the other hand, has been proposed as a highly promising drug candidate for a wide variety of CNS disorders. While this study might be of great interest to readers, it needs major revision before publication.

First, the title should be revised because in the current form it points toward a clinical study conducted with CBD. In contrast, the study was performed on whole blood culture.

@ANSWER: the title is changed into:

In vitro effects of cannabidiol on activated immune-inflammatory pathways in major depressive patients and healthy controls.

Second, the abstract should be rewritten:
- Methods, Results, Conclusion words should be omitted

@ANSWER: “Methods, Results, Conclusion” are deleted 

- the Results part should be rephrased because the observed increase of Th-1, Th-2 etc. profile has been induced by lipopolysaccharide + phytohaemagglutinin, and is not an inherent characteristic of depression.

@ANSWER. This is rephrased as:

 Depression was characterized by significantly increased PHA+LPS-stimulated Th-1, Th-2, Th-17, T …..

- Conclusion: there was no beneficial effect of CBD on the activated immune profile - the authors should emphasize this, and not speculate "very complex". And again, this should not be generalized - please specify that in this experimental context.

@ANSWER: this sentence is changed into

In summary, in this experiment, there was no beneficial effect of CBD on the activated immune profile of depression and higher CBD concentrations can worsen inflammatory processes.

Third, in the Introduction section, more evidence exists that CBD might have antidepressant-like effects. In lines 82-84, please cite 10.3390/biom10050801 and 10.1016/j.pnpbp.2021.110508 to have more updated reference list.

@ANSWER: we have added these two papers:

Gáll Z, Farkas S, Albert Á, Ferencz E, Vancea S, Urkon M, Kolcsár M. Effects of Chronic Cannabidiol Treatment in the Rat Chronic Unpredictable Mild Stress Model of Depression. Biomolecules. 2020 May 22;10(5):801. doi: 10.3390/biom10050801. PMID: 32455953; PMCID: PMC7277553.

Martín-Sánchez A, González-Pardo H, Alegre-Zurano L, Castro-Zavala A, López-Taboada I, Valverde O, Conejo NM. Early-life stress induces emotional and molecular alterations in female mice that are partially reversed by cannabidiol. Prog Neuropsychopharmacol Biol Psychiatry. 2022 Apr 20;115:110508. doi: 10.1016/j.pnpbp.2021.110508. Epub 2021 Dec 30. PMID: 34973413.

Fourth, in the Methods section:
The authors did not set any criteria for BMI when selecting patients for the study. Moreover, in the Results section, they showed that the depressive group has had a significantly higher BMI. However, there is plenty of evidence that obesity can be linked with a low-grade chronic inflammation characterized by macrophage activation and TNFa production. It is stated that this confounding was controlled for in the statistical analysis phase, but it might be worth commenting on this.

ANSWER: Now addressed in the limitation section as:

Third, the BMI was higher in depressed patients and it is known that increasing BMI is accompanied by increases in inflammatory biomarkers (Cohen et al., 2020). The differences in inflammatory biomarkers between people with a normal BMI and those with overweight are very small, although significant and detectable in large scale studies (Cohen et al., 2020). Nevertheless, in the present study we statistically controlled for any effects of BMI (and age and sex) and found that all results shown in this study did not change after adjusting for BMI, while BMI had no significant effects. By inference, depression explains a much greater part of the variance in the immune tests than BMI and the effects of the latter may not always be detected in smaller n studies.

Minor comments:
Results

Table 4 can be compacted by eliminating the df column and including the data in the text of the table caption.

@ANSWER: the df column is deleted and df shown in the legends to the Figure

Tables 6 and 7 should be formatted according to the journal's guidelines. 

@ANSWER: now well formatted

Discussion

Line 440 - This reference is more relevant: 10.3390/ijms23010094

@ANSWER: we have added this reference

Line 585 - Please check.

@ANSWER:I changed this sentence into:

Third, administration of CBD induced sustained antidepressant effects which are associated with changes in synaptic plasticity in the prefrontal cortex through stimulation of the BDNF signaling pathway [71].

Round 2

Reviewer 1 Report

The authors have taken into account my remarks. So the manuscript ca
n be accepted in its present form.

Reviewer 2 Report

All of the reviewer's concerns have been addressed by the authors.